

# An efficient frequency-domain model for quick load analysis of floating offshore wind turbines

Antonio Pegalajar-Jurado[1], Michael Borg[1], and Henrik Bredmose[1]

[1]Department of Wind Energy, Technical University of Denmark, Nils Koppels Allé 403, DK-2800 Kongens Lyngby, Denmark

**Correspondence:** Antonio Pegalajar-Jurado (ampj@dtu.dk)

**Abstract.** A model for Quick Load Analysis of Floating wind turbines, QuLAF, is presented and validated here. The model is a linear, frequency-domain, efficient tool with four planar degrees of freedom: platform surge, heave, pitch and tower modal deflection. The model relies on state-of-the-art tools from which hydrodynamic, aerodynamic and mooring loads are extracted and cascaded into QuLAF. Hydrodynamic and aerodynamic loads are precomputed in WAMIT and FAST respectively, while the mooring system is linearized around the equilibrium position for each wind speed using MoorDyn. An approximate approach to viscous hydrodynamic damping is developed, and the aerodynamic damping is extracted from decay tests specific for each degree of freedom. Without any calibration, the model predicts the motions of the system in stochastic wind and waves with good accuracy when compared to FAST. The damage-equivalent bending moment at the tower bottom is estimated with errors between 0.2% and 11.3% for all the load cases considered. The largest errors are associated with the most severe wave climates for wave-only conditions and with turbine operation around rated wind speed for combined wind and waves. The computational speed of the model is between 1300 and 2700 times faster than real-time.

## 1 Introduction: the need for an efficient, frequency-domain tool

Offshore wind energy is a key contributor to a carbon-free energy supply. Most of today's offshore wind farms are bottom-fixed, meaning their feasibility is limited to shallow and intermediate water depths. On the other hand, the wind resource in deep water represents an enormous potential that can be unlocked with the deployment of floating wind farms. An important step in making floating wind turbines economically feasible is the application of larger wind turbines and the ability to design the floater to a minimum cost. The design of a floating platform for offshore wind deployment depends on many design variables, and each possible combination of design variables is a potential design. In the design process, the candidate designs need to be simulated in different environmental conditions in order to assess the magnitude of the motions and loads in the system. These simulations are typically carried out with time-domain numerical tools, which allow a representative modelling of the physical phenomena involved, and can simulate at about real-time CPU speed. However, these models can be computationally expensive, especially if one needs to evaluate different floater designs under several environmental conditions. For an improved design process, faster tools are needed to allow optimization in the initial design stage, where the design space has to be thoroughly explored and a broad overview of the system response is desirable.



A few studies of simplified design models for offshore wind turbine floaters exist in the literature. Lupton (2014) presented a frequency-domain numerical tool for the analysis of the OC3-Hywind spar floating wind turbine (Jonkman, 2010), with eight degrees of freedom (DoFs): one normal mode per blade, two tower fore-aft modes, and platform surge, heave and pitch. The model included linear hydrodynamics computed with a potential-flow panel code and linearized viscous drag. The aero-
dynamic forces were included through harmonic linearization, and the mooring lines were represented by a stiffness matrix. The frequency-domain code was benchmarked against an equivalent Bladed (DNV-GL AS, 2016) model with Morison-based hydrodynamics, and with a stiffness mooring matrix. Neither the frequency-domain model nor the Bladed model included viscous drag. Results were shown for regular waves and uniform, harmonic wind, and the frequency-domain code was reported to be up to 37 times faster than the Bladed model. In Lemmer et al. (2016) a simplified time-domain model of the
OC3-Hywind spar (Jonkman, 2010) and OC4-DeepCwind semi submersible (Robertson et al., 2014) floating wind turbines was introduced. The model has four DoFs: platform surge and pitch, tower first fore-aft mode, and rotor azimuthal position. Linear hydrodynamics from a radiation-diffraction panel code was included in the time-domain model through the Cummins equation (Cummins, 1962). Aerodynamics was computed by coupling the code to AeroDyn. Quasi-static mooring forces were computed by solving the catenary mooring equations at each time step. A linearized version of the code was also presented. In
the results, the linearized frequency-domain version was successfully benchmarked against the nonlinear time-domain version, by comparing the linear transfer function from wave height to tower-top displacement to its nonlinear equivalent. The work of Wang et al. (2016) involved a frequency-domain model of the DeepCwind semi submersible (Robertson et al., 2014) with two rigid-body DoFs: platform surge and pitch. Linear hydrodynamics, linearized drag and drift forces were computed with the commercial software AQWA. The aerodynamic loads were included through a linearized version of the actuator disk equation,
where the aerodynamic contribution was divided into a constant force and a damping term — thus neglecting stochastic wind forcing. The mooring loads were included through a stiffness matrix, obtained from both quasi-static and dynamic mooring models. The model was validated against DeepCwind test data in terms of natural frequencies, response-amplitude operators (RAOs) and power spectral density (PSD) plots of surge and pitch response, generally obtaining a good agreement. However, a frequency-domain model for floating wind turbines able to incorporate realistic aerodynamic loads is still needed.

For bottom-fixed offshore wind turbines, Schløer et al. (2018) recently developed a quick, frequency-domain model named QuLA (**Qu**ick **L**oad **A**nalysis), using the DTU 10MW Reference Wind Turbine (RWT) (Bak et al., 2013). The foundation and the wind turbine tower were defined as an Euler beam, and the first fore-aft modal deflection of this beam was the only DoF. The rotor and nacelle were represented by a point mass at the tower top, and aerodynamic forces and damping were precomputed in the time-domain aeroelastic tool Flex5 (Øye, 1996). Hydrodynamic forcing was included through the Morison
equation (Morison et al., 1950), where the structure velocity and acceleration were neglected. The code was validated against Flex5 in terms of time series, PSD, exceedance probability curves and fatigue damage-equivalent load (DEL). The bending moment at the seabed was estimated by QuLA within a 5% error, and the code was reported to be approximately 40 times faster than its Flex5 equivalent.

This study presents the extension of QuLA to floating offshore wind turbines. The resulting model, QuLAF (**Qu**ick **L**oad
**A**nalysis **F**loating), was first presented in Pegalajar-Jurado et al. (2016), with only two DoFs: floater surge and tower first





fore-aft bending mode. Here we present an improved version of the model, a frequency-domain code that captures the four dominant DoFs in the in-plane global motion: floater surge, heave and pitch, and tower first fore-aft modal deflection. The model, which is here adapted to the DTU 10MW RWT mounted on the OO-Star Wind Floater Semi 10MW platform (Yu et al., 2018), was set up through cascading techniques. Here, information is precomputed or extracted from more advanced models

(*parent* models) to enhance the simplified models (*children* models). In this case, the hydrodynamic loads are extracted from the radiation-diffraction, potential-flow solver WAMIT (Lee and Newman, 2016). The aerodynamic loads and aerodynamic damping are precomputed in the numerical tool FAST v8 (Jonkman and Jonkman, 2016), and the mooring module MoorDyn (Hall, 2016) is employed to extract a mooring stiffness matrix for different operating positions. This way, the model includes standard radiation-diffraction theory and realistic rotor loads through precomputed aeroelastic simulations. In the model, the

system response is obtained by solving the linear equations of motion (EoM) in the frequency domain, leading to a very efficient tool. While the radiation-diffraction results allow a full linear response evaluation, the ambition of this model is to extend them with the flexible tower and realistic rotor loads. The results from QuLAF are here benchmarked against results from its time-domain, state-of-the-art (SoA) *parent* model, in terms of time series, PSD, exceedance probability and fatigue DEL. We are able to assess the strengths and weaknesses of the cascading process, and further develop techniques to improve

the accuracy of the simplified model. In this way, the potential of the model as a reliable tool for pre-design is demonstrated. The idea is that more advanced SoA models can be used in the analysis of load cases once the conceptual floater design is established with the efficient pre-design model.

## 2 The case study

The floating wind turbine chosen for the present study is the DTU 10MW Reference Wind Turbine (Bak et al., 2013) mounted

on the OO-Star Wind Floater Semi 10MW platform (Yu et al., 2018). The main properties of the DTU 10MW RWT are given in Tab. 1 below, and further information can be found in Bak et al. (2013). The Basic DTU Wind Energy controller (Hansen and Henriksen, 2013) is utilized, tuned to avoid the platform pitch instability commonly known as the "negative damping problem" reported in, for example, Larsen and Hanson (2007).

**Table 1.** Key figures for the DTU 10MW Reference Wind Turbine.

| Rated power | Rated wind speed | Wind regime | Rotor diameter | Hub height |
|---|---|---|---|---|
| 10 MW | 11.4 m/s | IEC Class 1A | 178.3 m | 119 m |

The floating platform (see Fig. 1), developed by Dr. techn. Olav Olsen (www.olavolsen.no), is a semi submersible floater

made of post-tensioned concrete. It has a central column and three outer columns mounted on a star-shaped pontoon with three legs. Each outer column is connected to the sea bed by a catenary mooring line with a suspended clump weight. The main properties of the floating platform are collected in Tab. 2, and further information can be found in Yu et al. (2018).



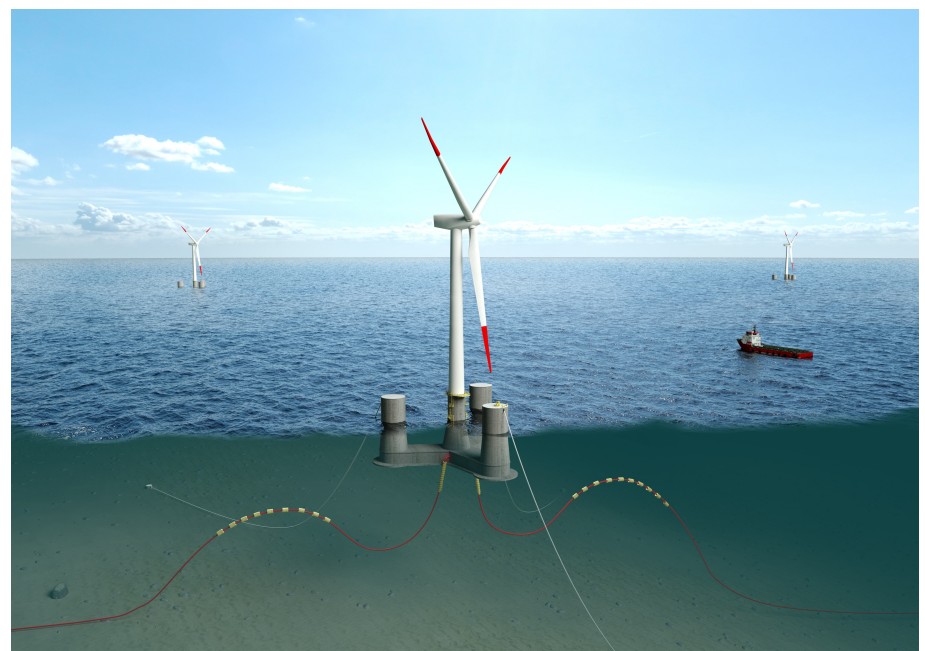

**Figure 1.** The OO-Star Wind Floater Semi 10MW concept (www.olavolsen.no).

**Table 2.** Key figures for the OO-Star Wind Floater Semi 10MW platform.

| Water depth | Mooring length | Draft | Freeboard | Displaced volume | Mass incl. ballast |
|---|---|---|---|---|---|
| 130 m | 703 m | 22 m | 11 m | 23509 $m^3$ | 21709 t |

## 3 Time vs. frequency domain: advantages and disadvantages

Floating wind turbines can be considered harmonic oscillators with multiple, coupled DoFs. To illustrate the strengths and weaknesses of solving the relevant EoM in the time or frequency domain, a simple one-DoF mass-spring-damper system is considered,

$$m\ddot{x}(t) + b\dot{x}(t) + cx(t) = F(t), \tag{1}$$

where $m$ is the system mass, $b$ is the damping coefficient, $c$ is the restoring coefficient, $x(t)$ is the system displacement from its equilibrium position, and $F(t)$ is a harmonic excitation force. Equation (1) can be also written in complex notation, by expressing the excitation force at the frequency $\omega$ as $F(t) = \Re\{\hat{F}(\omega)e^{i\omega t}\}$, where $\Re\{\}$ indicates the real part, $\hat{F}(\omega)$ is the Fourier transform of $F(t)$ and $i$ is the imaginary unit. If the transient part of the response is neglected, the steady-state system





response at the given frequency can also be written as $x(t) = \Re\{\hat{x}(\omega)e^{i\omega t}\}$, leading to the equation of motion in the frequency domain,

$$[-\omega^2 m + i\omega b + c]\hat{x}(\omega) = \hat{F}(\omega) \quad \Longrightarrow \quad \hat{x}(\omega) = \frac{\hat{F}(\omega)}{-\omega^2 m + i\omega b + c} \equiv H(\omega)\hat{F}(\omega). \tag{2}$$

The frequency-domain response $\hat{x}(\omega)$ can be obtained by simply multiplying the frequency-domain excitation force $\hat{F}(\omega)$
by the transfer function $H(\omega)$. This can be done at all frequencies and, due to the linearity, one can add the results at each frequency to get the total solution. Thus, once $\hat{x}(\omega)$ has been determined for all frequencies, the time-domain response $x(t)$ is obtained through an inverse Fourier transform of $\hat{x}(\omega)$. If fast Fourier transform (FFT) and fast inverse Fourier transform (iFFT) are used, the solution can be obtained very quickly. Figure 2 shows the response $x(t)$ of a one-DoF oscillator subjected to stochastic hydrodynamic linear forcing. The response labeled as "Time domain" was obtained by time-stepping of Eq.
(1) with the classical $4^{th}$-order Runge-Kutta method and initial conditions $x(0) = 0$ and $\dot{x}(0) = 0$. The response labeled as "Frequency domain" was computed by first obtaining the frequency-domain excitation force $\hat{F}(\omega) = \text{FFT}(F(t))$, calculating the frequency-domain response using Eq. (2), and finally going back to the time-domain response, $x(t) = \Re\{\text{iFFT}(\hat{x}(\omega))\}$. The simulation time step was 0.01 s and the total simulated time was 600 s, although only the first 60 s are shown here. The time-domain solution took 13.074 s to run, while the frequency-domain solution was done in 0.005 s, or 2615 times faster.
The two responses diverge at the beginning, where the time-domain solution is dominated by the transient response, which is neglected in the frequency-domain solution. However, after approximately 30 s and until the end of the simulation, the two solutions are practically identical.

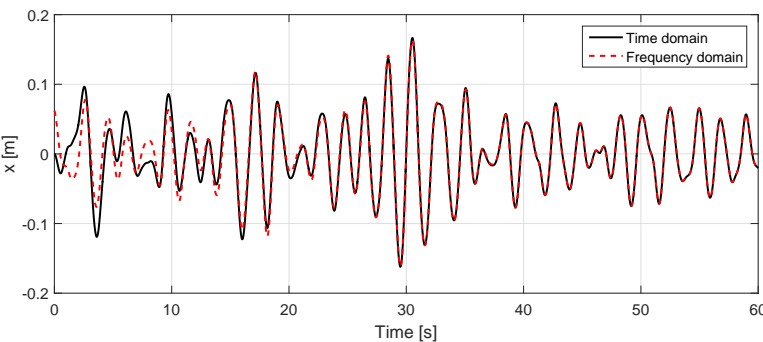

**Figure 2.** Response of a one-DOF, mass-spring-damper system to stochastic hydrodynamic linear forcing. Only the first 60 s are shown.

In addition to the gain in CPU speed, solving the EoM in the frequency domain allows an easier handling of frequency-dependent properties, such as hydrodynamic added mass and radiation damping. On the other hand, it has also been shown that
transient effects are only captured by time-domain models. Perhaps the most clear disadvantage of frequency-domain models is that they can only accomodate loads that depend linearly on the response and its time derivatives, such as hydrodynamic added mass loads or hydrostatic loads. They cannot directly accomodate loads that depend on the response in a nonlinear manner,





such as viscous drag from relative structural motion or catenary mooring loads. Here, simplified or linearized formulations have to be implemented instead.

## 4    The time-domain, state-of-the-art numerical model

In state-of-the-art models the nacelle, hub and floating platform are often considered rigid, whereas the tower and blades are
flexible. The platform motion typically has six DoFs: surge, sway, heave, roll, pitch and yaw. Aerodynamics are normally computed using unsteady Blade Element Momentum (BEM) theory (Hansen, 2008). Hydrodynamics are typically represented by radiation-diffraction theory (Newman, 1980), the Morison equation, or a combination of both. The mooring lines can be modelled with either quasi-static or dynamic approaches. In general, SoA models are more accurate than simplified models, but they also have a higher CPU cost.

A state-of-the-art, time-domain numerical model of the OO-Star Semi + DTU 10MW floating wind turbine was used in this study as a parent model to QuLAF. The SoA model was implemented in FAST v8.16.00a-bjj (Jonkman and Jonkman, 2016) with active control and 15 DoFs for turbine and platform: first and second flapwise blade modal deflections, first edgewise blade modal deflection, drivetrain rotational flexibility, drivetrain speed, first and second fore-aft and side-side tower modal deflections, and platform surge, sway, heave, roll, pitch and yaw. The turbulent wind fields were computed in TurbSim, and the
aerodynamics loads were modelled with AeroDyn v14. The Basic DTU Wind Energy controller interacts with FAST through a dynamic link library (DLL). The mooring loads, calculated by MoorDyn (Hall, 2016), include buoyancy, mass inertia and hydrodynamic loads resulting from the motion of the mooring lines in calm water. Hydrodynamic loads on the platform were first computed in WAMIT (Lee and Newman, 2016) and are coupled to FAST through the Cummins equation. Viscous effects were modelled internally by the Morison drag term. Further details on the modelling of floating wind turbines in FAST can be
found in Jonkman (2009), while a thorough description of the FAST model used in this study is presented in Pegalajar-Jurado et al. (2018a) and Pegalajar-Jurado et al. (2018b).

## 5    The frequency-domain, cascaded numerical model

The simplest model for the dynamic analysis of floating wind turbines would only have a few DoFs, typically rigid-body motion of the platform in surge and pitch. Aerodynamic loads would be represented by a point force at the rotor hub and
defined by an actuator disk model. If the floating platform is slender compared to the incident waves, a strip-theory approach can be applied to compute the hydrodynamic loads from the Morison equation. The forces exerted by the mooring system can be included through a stiffness matrix in the linear equation of motion. Simplified, low-order models are very CPU-efficient but their accuracy is often limited. The simplified model presented in this paper combines elements extracted from a SoA model into a very efficient tool, and aims at getting close to the accuracy of the SoA model, while still retaining the CPU efficiency
of low-order models.



QuLAF represents the floating wind turbine as two lumped masses — floating platform and rotor-nacelle assembly — connected by a flexible tower. The model captures four DoFs: platform surge, heave, pitch and tower fore-aft modal deflection. The floating wind turbine used for this study is represented in QuLAF as depicted in Fig. 3. The equation of motion is a matrix version of Eq. (2),

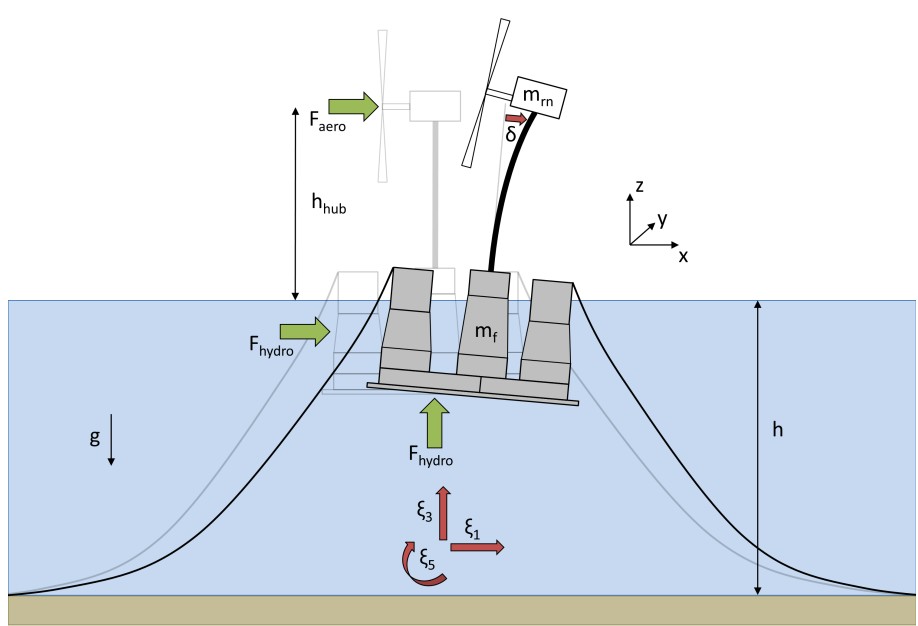

**Figure 3.** Sketch of the floating wind turbine as seen by the QuLAF model.

$$5 \quad \left[-\omega^2(\mathbf{M} + \mathbf{A}(\omega)) + i\omega\mathbf{B}(\omega) + \mathbf{C}\right]\hat{\boldsymbol{\xi}}(\omega) = \hat{\mathbf{F}}(\omega) \implies \hat{\boldsymbol{\xi}}(\omega) = \mathbf{H}(\omega)\hat{\mathbf{F}}(\omega), \tag{3}$$

where $\mathbf{M}$ is the structural mass and inertia matrix, $\mathbf{A}(\omega)$ is the frequency-dependent, hydrodynamic added mass and inertia matrix, $\mathbf{B}(\omega)$ is the frequency-dependent damping matrix and $\mathbf{C}$ is the restoring matrix. The vector $\hat{\boldsymbol{\xi}}(\omega)$ is the response in the frequency domain for the four degrees of freedom and $\hat{\mathbf{F}}(\omega)$ is the vector of excitation forces and moments in the frequency domain. The system transfer function is given by $\mathbf{H}(\omega)$. The different elements in Eq. (3) are described in detail below.




## 5.1 Structural mass and inertia matrix

The matrix of structural mass and inertia is defined as

$$
\mathbf{M} = \begin{bmatrix}
m_{tot} & 0 & m_{tot} z_{tot}^{CM} & m_{rn}\phi_{hub} + \sum_{i=1}^{N_t} \tilde{\rho}_i \phi_i \Delta z_i \\
 & m_{tot} & 0 & 0 \\
 & & I_{tot}^O & m_{rn}\phi_{hub}h_{hub} + I_{rn}^{TT}\phi_{z,hub} + \sum_{i=1}^{N_t} \tilde{\rho}_i \phi_i z_i \Delta z_i \\
 & & & m_{rn}\phi_{hub}^2 + I_{rn}^{TT}\phi_{z,hub}^2 + \sum_{i=1}^{N_t} \tilde{\rho}_i \phi_i^2 \Delta z_i
\end{bmatrix},
\tag{4}
$$

where $m_{tot}$ is the total mass of the floating wind turbine, $m_{tot} = m_f + m_{rn} + \sum_{i=1}^{N_t} \tilde{\rho}_i \Delta z_i$, which includes the mass of the
floater $m_f$, the rotor-nacelle mass $m_{rn}$ and the mass sum of all the $N_t$ elements that compose the flexible tower, each with a
mass per length $\tilde{\rho}_i$ and a height $\Delta z_i$. The total mass inertia of the system around the flotation point $O$ is given by $I_{tot}^O = I_f^O +$
$I_{rn}^O + \sum_{i=1}^{N_t} \tilde{\rho}_i z_i^2 \Delta z_i$, including the floater inertia $I_f^O$, the rotor-nacelle inertia $I_{rn}^O$ and the inertia of each of the tower elements,
located at an absolute height $z_i = (z_{t,i} + h_t)$, where $z_{t,i}$ is the height of the element $i$ with respect to the tower base, located at
a height $h_t$. The centre of mass (CM) of the whole structure is given by $z_{tot}^{CM} = (m_f z_f^{CM} + m_{rn}h_{hub} + \sum_{i=1}^{N_t} \tilde{\rho}_i z_i \Delta z_i)/m_{tot}$,
with contributions from the floater CM $z_f^{CM}$, the rotor-nacelle CM at the hub height $h_{hub}$, and the CM of each of the tower
elements. The mode shape deflection of the tower evaluated at a generic tower element $i$ is $\phi_i$, while $\phi_{hub}$ and $\phi_{z,hub}$ are the
mode shape deflection and its slope evaluated at the hub. Finally, $I_{rn}^{TT}$ represents the mass inertia of the rotor-nacelle assembly
referred to the tower top. The tower structural properties and first mode shape are the same as the ones given as an input to the
state-of-the-art model.

## 5.2 Hydrodynamic added mass matrix and damping matrix

The frequency-dependent, hydrodynamic added mass and radiation damping matrices, $\mathbf{A}(\omega)$ and $\mathbf{B}_{rad}(\omega)$, can be precomputed
in a radiation-diffraction solver. Here, the same WAMIT output files used for the SoA model are loaded into QuLAF. However,
the original 6x6 matrices are reduced by removing the rows and columns corresponding to the DoFs not included in the
simplified model (sway, roll, yaw), and a row and column of zeros is added for compatibility with the tower DoF,

$$
\mathbf{A}(\omega) = \begin{bmatrix}
a_{11}(\omega) & a_{13}(\omega) & a_{15}(\omega) & 0 \\
a_{31}(\omega) & a_{33}(\omega) & a_{35}(\omega) & 0 \\
a_{51}(\omega) & a_{53}(\omega) & a_{55}(\omega) & 0 \\
0 & 0 & 0 & 0
\end{bmatrix}
\qquad
\mathbf{B}_{rad}(\omega) = \begin{bmatrix}
b_{11}(\omega) & b_{13}(\omega) & b_{15}(\omega) & 0 \\
b_{31}(\omega) & b_{33}(\omega) & b_{35}(\omega) & 0 \\
b_{51}(\omega) & b_{53}(\omega) & b_{55}(\omega) & 0 \\
& 0 & 0 & 0
\end{bmatrix}.
\tag{5}
$$





The global damping matrix includes contributions from the hydrodynamic radiation damping $\mathbf{B}_{rad}(\omega)$, the hydrodynamic viscous damping $\mathbf{B}_{vis}$, the aerodynamic damping $\mathbf{B}_{aero}(\omega)$ and the tower structural damping $\mathbf{B}_{struc}$,

$$\mathbf{B}(\omega) = \mathbf{B}_{rad}(\omega) + \mathbf{B}_{vis} + \mathbf{B}_{aero}(\omega) + \mathbf{B}_{struc}. \tag{6}$$

The hydrodynamic viscous damping matrix $\mathbf{B}_{vis}$ is analytically extracted from the Morison equation, as shown in section

5.8.1. The aerodynamic damping matrix,

$$\mathbf{B}_{aero}(\omega) = \begin{bmatrix} b_{aero,11}(\omega) & 0 & 0 & 0 \\ & 0 & 0 & 0 \\ & & b_{aero,55}(\omega) & 0 \\ & & & b_{aero,tow} \end{bmatrix}, \tag{7}$$

is extracted from the SoA model for each mean wind speed $W$, as detailed in section 5.8.2. The matrix of structural damping is given by

$$\mathbf{B}_{struc} = \begin{bmatrix} 0 & 0 & 0 & 0 \\ & 0 & 0 & 0 \\ & & 0 & 0 \\ & & & 2\zeta_{struc,tow}\sqrt{C_{tow}M_{tow}} \end{bmatrix}, \tag{8}$$

where the structural damping ratio for the first fore-aft tower mode, $\zeta_{struc,tow}$, is directly taken from the input to the state-of-the-art model, and $C_{tow}$ and $M_{tow}$ are the last diagonal elements of the system restoring matrix $\mathbf{C}$ and the mass inertia matrix $\mathbf{M}$, respectively.

## 5.3 Restoring matrix

The restoring matrix includes hydrostatic stiffness $\mathbf{C}_{hst}$, structural stiffness $\mathbf{C}_{struc}$ and mooring stiffness $\mathbf{C}_{moor}$,

$$\mathbf{C} = \mathbf{C}_{hst} + \mathbf{C}_{struc} + \mathbf{C}_{moor}. \tag{9}$$



The hydrostatic matrix should only include the contributions from centre of buoyancy (CB) and waterplane area. It is computed as part of the radiation-diffraction solution, and is reduced following the same procedure as for the added mass and radiation damping matrices. The structural stiffness matrix is given by

$$
\mathbf{C}_{struc} =
\begin{bmatrix}
& 0 & 0 & 0 \\
 & 0 & 0 & 0 \\
 & & -m_{tot}g z_{tot}^{CM} & -m_{rn}g\phi_{hub} - \sum\limits_{i=1}^{N_t} \tilde{\rho}_i g\phi_i \Delta z_i \\
 & & & \sum\limits_{i=1}^{N_t} EI_i \phi_{zz,i}^2 \Delta z_i
\end{bmatrix},
\tag{10}
$$

where $g$ is the acceleration of gravity, and $EI_i$ and $\phi_{zz,i}$ are the bending stiffness and the curvature of the mode shape for the tower element $i$, respectively. The off-diagonal term represents the negative restoring effect of the tower and rotor-nacelle mass on the tower DoF when the platform pitches. The mooring restoring matrix $\mathbf{C}_{moor}$ is extracted from the SoA model for each mean wind speed $W$, as detailed in section 5.8.3.

### 5.4   Dynamic response vector

The dynamic response vector,

$$
\hat{\boldsymbol{\xi}}(\omega) =
\begin{bmatrix}
\hat{\xi}_1(\omega) \\
\hat{\xi}_3(\omega) \\
\hat{\xi}_5(\omega) \\
\hat{\alpha}(\omega)
\end{bmatrix},
\tag{11}
$$

has one element for each DoF: platform surge, heave, pitch and tower fore-aft modal deflection. The sign convention is that shown in Fig. 3, with positive surge in the downwind direction, positive heave upwards, positive pitch (about flotation point) clockwise and positive tower deflection in the downwind direction. The physical tower deflection at any height $z$ can

be obtained by multiplying the mode shape and the modal deflection, $u(z,t) = \phi(z)\alpha(t)$. The tower-top deflection is therefore given by $\delta(t) = u(h_{hub},t) = \phi_{hub}\alpha(t)$. If the absolute nacelle displacement is sought, the contributions from platform surge and pitch motion must be added to the tower deflection, and the global response vector $\hat{\boldsymbol{\xi}}_{glob}(\omega)$ is found by introducing a transformation matrix $\mathbf{T}_{glob}$,

$$
\hat{\boldsymbol{\xi}}_{glob}(\omega) =
\begin{bmatrix}
1 & 0 & 0 & 0 \\
0 & 1 & 0 & 0 \\
0 & 0 & 1 & 0 \\
& 0 & h_{hub} & \phi_{hub}
\end{bmatrix}
\begin{bmatrix}
\hat{\xi}_1(\omega) \\
\hat{\xi}_3(\omega) \\
\hat{\xi}_5(\omega) \\
\hat{\alpha}(\omega)
\end{bmatrix}
= \mathbf{T}_{glob}\hat{\boldsymbol{\xi}}(\omega).
\tag{12}
$$





## 5.5 Dynamic load vector

The dynamic load vector,

$$\hat{\mathbf{F}}(\omega) = \hat{\mathbf{F}}_{hydro}(\omega) + \hat{\mathbf{F}}_{aero}(\omega), \tag{13}$$

contains hydrodynamic loads $\hat{\mathbf{F}}_{hydro}(\omega)$ and aerodynamic loads $\hat{\mathbf{F}}_{aero}(\omega)$. Hydrodynamic loads are extracted from the
solution to the diffraction problem, which provides a vector of wave excitation forces and moments in all six degrees of
freedom, namely $\hat{\mathbf{X}}(\omega)$. These excitation forces are normalized to waves of unit amplitude, therefore the wave loads for a
specific time series of free-surface elevation $\eta(t)$ are obtained by multiplying $\hat{\mathbf{X}}(\omega)\hat{\eta}(\omega)$. The vector of wave excitation forces
and moments is also reduced to adapt it to the simplified model,

$$\hat{\mathbf{F}}_{hydro}(\omega) = \hat{\mathbf{X}}(\omega)\hat{\eta}(\omega) \equiv \begin{bmatrix} \hat{X}_1(\omega) \\ \hat{X}_3(\omega) \\ \hat{X}_5(\omega) \\ 0 \end{bmatrix} \hat{\eta}(\omega), \tag{14}$$

where $\hat{\eta}(\omega)$ can be computed both from an input time series $\eta(t)$ or from a theoretical wave spectrum. The only viscous
effect considered in the model is viscous damping (see section 5.8.1), but viscous forcing is neglected to keep the model
computationally efficient. This simplification, however, is considered reasonable because hydrodynamics for this platform is
dominated by inertia loads, and viscous forcing is expected to be relevant mainly for severe sea states, which lie on the border
of the model's applicability. The vector of aerodynamic loads contains only the dynamic part of the wind loads and has the
format

$$\hat{\mathbf{F}}_{aero}(\omega) = \begin{bmatrix} \hat{F}_{aero,1}(\omega) \\ \hat{F}_{aero,3}(\omega) \\ \hat{F}_{aero,1}(\omega)h_{hub} + \hat{\tau}_{aero}(\omega) \\ \hat{F}_{aero,1}(\omega)\phi_{hub} + \hat{\tau}_{aero}(\omega)\phi_{z,hub} \end{bmatrix}, \tag{15}$$

where $\hat{F}_{aero,1}(\omega)$ and $\hat{F}_{aero,3}(\omega)$ represent the horizontal and vertical components of the aerodynamic loads on the rotor,
respectively. The aerodynamic tilt torque on the rotor is given by $\hat{\tau}_{aero}(\omega)$. The time-domain aerodynamic loads for each mean
wind speed $W$ are precomputed in the SoA model, as detailed in section 5.8.2.





### 5.6 Static load and response

Static loads are related to the equilibrium of the structure. In the model, the static part of the response, $\boldsymbol{\xi}_{st}$, is added to the dynamic part $\hat{\boldsymbol{\xi}}(\omega)$ when it is converted from frequency to time domain via iFFT. The static loads applied are

$$\mathbf{F}_{st} = \mathbf{F}_{aero,st} + \mathbf{F}_{grav} + \mathbf{F}_{buoy}, \tag{16}$$

5  which include the static part of the aerodynamic loads $\mathbf{F}_{aero,st}$, the gravity loads $\mathbf{F}_{grav}$ and the buoyancy loads $\mathbf{F}_{buoy}$. The gravity load vector is given by

$$\mathbf{F}_{grav} = \begin{bmatrix} 0 \\ -m_{tot}g - F_{moor,z} \\ m_{rn}g x_{rn}^{CM} \\ m_{rn}g x_{rn}^{CM} \phi_{z,hub} \end{bmatrix}, \tag{17}$$

where $F_{moor,z}$ is the vertical force exerted by the mooring lines in equilibrium, and $x_{rn}^{CM}$ is the horizontal coordinate of the rotor-nacelle CM. The buoyancy load vector is

$$10 \quad \mathbf{F}_{grav} = \begin{bmatrix} 0 \\ \rho_w g V_f \\ -\rho_w g V_f x_f^{CB} \\ 0 \end{bmatrix}, \tag{18}$$

where $\rho_w$ is the water density, $V_f$ is the volume displaced by the floating platform, and $x_f^{CB}$ is the horizontal coordinate of the floater CB.

The stiffness matrix $\mathbf{C}$ contains a contribution from the mooring system, $\mathbf{C}_{moor}$, which depends on the mean wind speed $W$. With no wind and only linear wave forcing, the floating wind turbine operates around its equilibrium position with a stiffness 15 $\mathbf{C}_0$. If wind is introduced, the floating wind turbine is moved to a new equilibrium position, where the stiffness matrix is $\mathbf{C}_W$. The static response is therefore obtained from the static loads by considering a mean stiffness matrix $\mathbf{C}_{st}$,

$$\mathbf{C}_{st} = \frac{\mathbf{C}_0 + \mathbf{C}_W}{2} \implies \mathbf{C}_{st}\boldsymbol{\xi}_{st} = \mathbf{F}_{st}. \tag{19}$$

This approximation is accurate to second-order.



## 5.7 System natural frequencies

The vector of natural frequencies $\boldsymbol{\omega_0}$ is found by solving the undamped eigenvalue problem given by

$$\left[-\boldsymbol{\omega_0}^2(\mathbf{M} + \mathbf{A}(\boldsymbol{\omega_0})) + \mathbf{C}\right]\hat{\boldsymbol{\xi}}(\boldsymbol{\omega_0}) = 0 \quad \implies \quad \boldsymbol{\omega_0^2}\hat{\boldsymbol{\xi}}(\boldsymbol{\omega_0}) = (\mathbf{M} + \mathbf{A}(\boldsymbol{\omega_0}))^{-1}\mathbf{C}\hat{\boldsymbol{\xi}}(\boldsymbol{\omega_0}). \tag{20}$$

Since the matrix of added mass depends on frequency, the eigenvalue problem is solved in a frequency loop. For each frequency $\omega$, the four possible natural frequencies are computed. When one of the four possible frequencies obtained is equal to the frequency of that particular iteration in the loop, a system natural frequency has been found. The system natural frequencies obtained with QuLAF are compared to those obtained with the SoA model in section 6.1.

## 5.8 Cascading techniques applied to the simplified model

In section 3, it was stated that one disadvantage of frequency-domain models is their inability to directly capture loads that depend on the response in a nonlinear way. Some obvious examples are viscous drag, aerodynamic loads and catenary mooring loads. This section gives a description of the cascading methods employed to incorporate such nonlinear loads into the simplified model.

### 5.8.1 Hydrodynamic viscous loads

Viscous effects on submerged bodies depend nonlinearly on the relative velocity between the wave particles and the structure, hence they can only be directly incorporated in time-domain models. In the offshore community this is normally done through the drag term of the Morison equation, which provides the transversal drag force $dF$ on a cylindrical member section of diameter $D$ and length $dl$ as

$$dF = \frac{1}{2}\rho C_D D|v_f - v_s|(v_f - v_s)dl, \tag{21}$$

where $\rho$ is the fluid density, $C_D$ is a drag coefficient, and $v_f$ and $v_s$ are the local fluid and structure velocities perpendicular to the member axis. The equation can be also written as

$$dF = \frac{1}{2}\rho C_D D \operatorname{sgn}(v_f - v_s)(v_f - v_s)^2 dl \equiv \frac{1}{2}\rho C_D D \operatorname{sgn}(v_f - v_s)(v_f^2 + v_s^2 - 2v_f v_s)dl, \tag{22}$$

which shows that the drag effects can be separated into a pure forcing term, a nonlinear damping term and a linear damping term. Since the hydrodynamics on the given floating platform are inertia-dominated and under the assumption of small displacements around the equilibrium position, the two first terms are neglected and only the linear damping term is retained



in the model. Invoking further the assumption of small displacements and velocities relative to the fluid velocity, we have $\mathrm{sgn}(v_f - v_s) \approx \mathrm{sgn}(v_f)$. With this assumption the linear damping term of the viscous force becomes

$$dF_l = \frac{1}{2}\rho C_D D \, \mathrm{sgn}(v_f - v_s)(-2v_f v_s)dl \approx -\rho C_D D |v_f| v_s dl. \tag{23}$$

A viscous damping matrix $\mathbf{B}_{vis}$ is now derived by applying Eq. (23) to the different DoFs. For the surge motion, integration

over the submerged body gives the total viscous force in the $x$ direction as

$$F_1 = -\int\limits_{z_{min}}^{0} \rho C_D D |u| \dot{\xi}_1 dz, \tag{24}$$

where $z_{min}$ is the structure's lowest submerged point, $u$ is the horizontal wave velocity and $\dot{\xi}_1$ is the surge velocity. The integral in Eq. (24) requires the estimation of drag coefficients and the computation of wave kinematics at several locations on the submerged structure, which can be involved for complex geometries. These computations would hinder on CPU efficiency,

so instead the local drag coefficient and wave velocity inside the integral are replaced by global, representative values outside the integral, $C_{Dx}$ and $u_{rep}$. Then the force becomes

$$F_1 = -\rho \dot{\xi}_1 \int\limits_{z_{min}}^{0} C_D D |u| dz \approx -\rho C_{Dx} u_{rep} \dot{\xi}_1 \int\limits_{z_{min}}^{0} D dz = -\rho C_{Dx} A_x u_{rep} \dot{\xi}_1 \equiv -b_{11}\dot{\xi}_1, \tag{25}$$

and the integral of the local diameter $D$ over depth is the floater's frontal area, $A_x$. This defines the surge-surge element of the viscous damping matrix. Further the $b_{51}$ element of the viscous damping matrix is obtained by consideration of the moment

from $F_1$ around the point of flotation,

$$\tau_1 = -\rho \dot{\xi}_1 \int\limits_{z_{min}}^{0} C_D D |u| z dz \approx -\rho C_{Dx} u_{rep} \dot{\xi}_1 \int\limits_{z_{min}}^{0} D z dz = -\rho C_{Dx} S_{y,Ax} u_{rep} \dot{\xi}_1 \equiv -b_{51}\dot{\xi}_1, \tag{26}$$

where $S_{y,Ax}$ is the first moment of area of $A_x$ about the flotation point and $b_{51}$ is the surge-pitch element of the viscous damping matrix. In a similar way, the heave-heave and heave-pitch coefficients of the viscous damping matrix are obtained by applying Eq. (23) to the heave motion,

$$F_3 = -\rho \dot{\xi}_3 \int\limits_{x_{min}}^{x_{max}} C_D D |w| dx \approx -\rho C_{Dz} w_{rep} \dot{\xi}_3 \int\limits_{x_{min}}^{x_{max}} D dx = -\rho C_{Dz} A_z w_{rep} \dot{\xi}_3 \equiv -b_{33}\dot{\xi}_3, \tag{27}$$

$$\tau_3 = \rho \dot{\xi}_3 \int\limits_{x_{min}}^{x_{max}} C_D D |w| x dx \approx \rho C_{Dz} w_{rep} \dot{\xi}_3 \int\limits_{x_{min}}^{x_{max}} D x dx = \rho C_{Dz} S_{y,Az} w_{rep} \dot{\xi}_3 \equiv -b_{53}\dot{\xi}_3. \tag{28}$$





Here $\dot{\xi}_3$ is the heave velocity, $w$ is the wave vertical velocity, $A_z$ is the floater's bottom area and $S_{y,Az}$ is the first moment of area of $A_z$ about the flotation point, which is zero for the present floating platform due to symmetry. Finally, by applying Eq. (23) to the pitch motion, the pitch-pitch element of the viscous damping matrix, $b_{55}$, is found. When the platform pitches with a velocity $\dot{\xi}_5$, a generic point at the floater with coordinates $(x, z)$ moves with a velocity $(z\dot{\xi}_5, -x\dot{\xi}_5)$. The motion creates
a moment due to viscous effects given by

$$\tau_5 = -\rho\dot{\xi}_5 \int_{z_{min}}^{0} C_D D |u| z^2 dz - \rho\dot{\xi}_5 \int_{x_{min}}^{x_{max}} C_D D |w| x^2 dx \approx -\rho(C_{Dx}I_{y,Ax}u_{rep} + C_{Dz}I_{y,Az}w_{rep})\dot{\xi}_5 \equiv -b_{55}\dot{\xi}_5, \qquad (29)$$

where $I_{y,Ax}$ and $I_{y,Az}$ are the second moments of area of $A_x$ and $A_z$, respectively. The complete matrix of viscous damping is therefore

$$\mathbf{B}_{vis} = \begin{bmatrix} \rho C_{Dx}A_x u_{rep} & 0 & \rho C_{Dx}S_{y,Ax}u_{rep} & 0 \\ & \rho C_{Dz}A_z w_{rep} & 0 & 0 \\ & & \rho(C_{Dx}I_{y,Ax}u_{rep} + C_{Dz}I_{y,Az}w_{rep}) & 0 \\ & & & 0 \end{bmatrix}. \qquad (30)$$

The global drag coefficients above have been chosen as $C_{Dx} = 1$ and $C_{Dz} = 2$, given that the bottom slab of the given floating platform has sharp corners and is expected to oppose a greater resistance to the flow (see Fig. 1). To obtain the representative velocity $u_{rep}$, the time- and depth-dependent horizontal wave velocity at the platform's centreline $u(0, z, t)$ is first averaged over depth and then over time,

$$u_{avg}(t) = \frac{1}{|z_{min}|} \int_{z_{min}}^{0} u(0, z, t)dz \equiv \frac{1}{|z_{min}|}\Re\left\{ \text{iFFT}\left( \frac{\omega\hat{\eta}(\omega)}{k}\left( 1 - \frac{\sinh(k(z_{min} + h))}{\sinh(kh)} \right) \right) \right\} \implies u_{rep} = \overline{|u_{avg}|}. \quad (31)$$

Here $k$ is the wave number for the angular frequency $\omega$ and $h$ is the water depth. The representative velocity $w_{rep}$ is chosen as the time average of the vertical wave velocity at the centre of the bottom plate,

$$w_{avg}(t) = w(0, z_{min}, t) \implies w_{rep} = \overline{|w_{avg}|}. \qquad (32)$$

This simplification of the wave field, although drastic, allows the characterization of the viscous damping for each sea state and avoids the need to compute wave kinematics locally and integrate the drag loads.

**5.8.2 Aerodynamic loads**

Aerodynamic loads depend on the square of the relative wind speed seen by the blades. The relative wind speed includes contributions from the rotor speed, the blade deflection, the tower deflection, and the motion of the platform. The fact that the





aerodynamic thrust depends on the blade relative velocity produces the well-known aerodynamic damping (e.g., Larsen and Hanson, 2007). State-of-the-art numerical models incorporate aerodynamic loads based on relative velocity, because both the wind speed and the blade structural velocity are known at each time step. However, the same cannot be done in a frequency-domain model. In the approach implemented in QuLAF, the aerodynamic loads considering the motion of the blades are

simplified and approximated by loads considering a fixed hub and linear damping terms. The time series of fixed-hub loads and the aerodynamic damping are extracted from the state-of-the-art model for each mean wind speed.

The aerodynamic loads are obtained at each wind speed $W$ by a SoA simulation with turbulent wind and no waves where all DoFs are disabled and the wind turbine controller is enabled. The time series of fixed-hub, pure aerodynamic loads $F_{aero,1}(t)$, $F_{aero,3}(t)$ and $\tau_{aero}(t)$ are extracted from the results and stored in a data file which is loaded into the model. Hence, these

FAST simulations need to be as long as the maximum simulation time needed in the simplified model (5400 s in this case).

For a given rotor, the work carried out by the aerodynamic damping is a function wind speed, rotational speed, turbulence intensity, motion frequency and oscillation amplitude. Here, we define an equivalent linear damping which delivers the same work over one oscillation cycle and can be extracted from a decay test. Schløer et al. (2018) used this principle for the tower fore-aft mode of a bottom-fixed offshore turbine and found that the damping was only slightly dependent of the motion am-

plitude. We make a further simplification and carry out the decay tests in steady wind. Since the mass and stiffness of floater and tower only affect the aerodynamic damping through the motion frequency, we transfer the damping coefficients $b$ from the decay tests in FAST to the QuLAF model. On the contrary, if the damping ratio $\zeta$ was transferred, changes in mass or stiffness properties would imply a change in the aerodynamic forcing, which is not physically correct. With the transfer of damping co-efficients $b$, re-calculation of the decay tests is only necessary in the event that the change of natural frequencies should affect

the damping values significantly. Here, the decay tests from which aerodynamic damping ratios are extracted were carried out at representative natural frequencies equal to those of the present floater. These decay tests in calm water and with the wind turbine controller active were carried out for each DoF with all the other DoFs locked. This way, the floating wind turbine is a one-DoF spring-mass-damper system, where the horizontal position of the hub $x_{hub}$ is of interest. The decay tests are carried out as a step test in steady wind where the wind speed goes from the minimum to the maximum value with step changes every

600 s. With every step change of wind speed, the structure moves to a new equilibrium position. If all sources of hydrodynamic and structural damping are disabled, the aerodynamic damping is the only responsible for the decay of the hub motion, and it can be extracted from the time series. The $n$ peaks extracted from the signal are used in pairs to estimate each local logarithmic decrement $d_i$, and from it, a local damping ratio $\zeta_i$, which is then averaged to obtain the aerodynamic damping ratio $\zeta_{aero}$ for the given DoF and $W$:

$$d_i = \log \frac{x_{hub,i}}{x_{hub,i+1}} \quad \implies \quad \zeta_i = \frac{d_i}{\sqrt{4\pi^2 + d_i^2}} \quad \implies \quad \zeta_{aero} = \frac{1}{n-1}\sum_1^{n-1}\zeta_i \tag{33}$$

Figure 4 shows examples of $x_{hub}(t)$ and selected peaks for surge, pitch and tower DoFs for a wind speed of 13 m/s. The wind changed from 12 m/s to 13 m/s at $t = 0$, and the mean of the signals has been subtracted. For surge and pitch, peaks within the first 40 s are neglected to allow the unsteady aerodynamic effects to disappear. For the tower DoF, however, the




frequency is much higher and the signal has died out by the time the aerodynamics are steady. For that reason, the tower decay peaks are extracted after 300 s, and a sudden impulse in wind speed is introduced at $t = 300$ s to excite the tower. This method was chosen since FAST does not allow an instantaneous force to be applied.

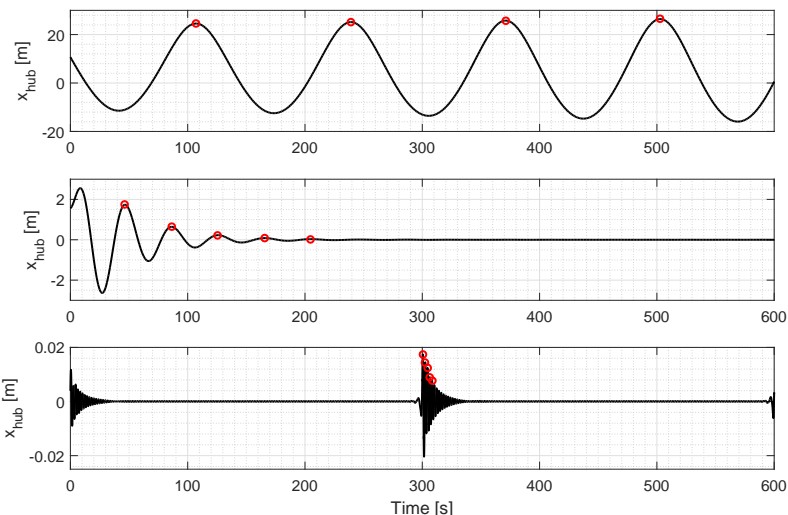

**Figure 4.** Example of time series of hub position and selected peaks for the extraction of aerodynamic damping. From top to bottom: surge, pitch and tower DoFs.

In Fig. 5 the aerodynamic damping ratio is shown for all DoFs as a function of $W$. It is observed that the aerodynamic damping in surge is negative for wind speeds between 11.4 m/s and 16 m/s, due to the wind turbine controller. However, in real environmental conditions with wind and waves, it has been observed that the hydrodynamic damping contributes to a positive global damping of the surge motion. This controller effect is similar to the "negative damping problem" reported in, for example, Larsen and Hanson (2007). The negative aerodynamic damping in surge may be eliminated if one tunes the controller natural frequency so it lies below the surge natural frequency of the floating wind turbine, as it was done in Larsen and Hanson (2007) for the floater pitch motion. This solution, however, would affect power production and was not adopted here because the global damping in surge has been observed to be positive when all other damping contributions are taken into account.

The damping ratio for each wind speed $\zeta_{aero,i}(W)$ is next converted to a damping coefficient by

$$b_{aero,i}(W) = 2\zeta_{aero,i}(W)\sqrt{C_{ii}(M_{ii} + A_{ii}(\omega))}, \tag{34}$$

where $C_{ii}$, $M_{ii}$ and $A_{ii}(\omega)$ are taken from the one-DoF oscillator in the corresponding decay test. The table of aerodynamic damping coefficients as a function of wind speed $b_{aero}(W)$ is stored in a data file, which is loaded into the model. Since the aerodynamic damping coefficients are extracted from simulations with steady wind, but applied in the model in simulations with turbulent wind, an averaging is applied to account for the variability of the wind speed in turbulent conditions. Given





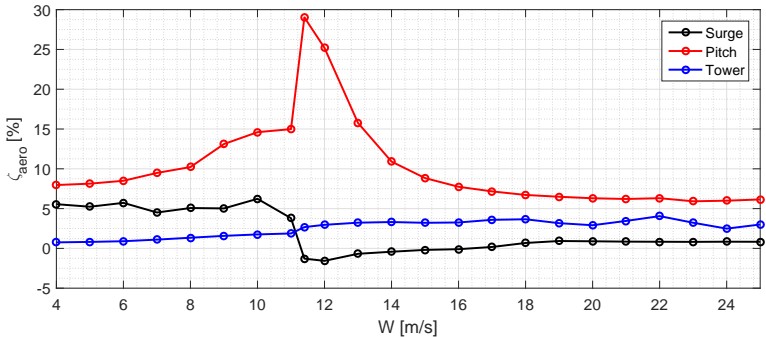

**Figure 5.** Aerodynamic damping ratios for different DoFs as a function of wind speed.

the time series of wind speed at hub height, $V(t)$, the probability density function (PDF) of a normal distribution given by $\mathcal{N}(\overline{V}, \sigma_V)$ is used to estimate the probability of occurrence within $V(t)$ of each discrete value of $W$. Then the aerodynamic coefficient for the given turbulent wind conditions and the $i^{th}$ DoF is

$$b_{aero,i} = \sum_{j=1}^{N_W} PDF(W_j)b_{aero,i}(W_j). \tag{35}$$

### 5.8.3 Mooring loads

The equations that provide the loads on a catenary cable depend nonlinearly on the fairlead position. In dynamic mooring models, the drag forces on the mooring cables are also included, therefore the mooring loads also depend on the square of the relative velocity seen by the lines. These nonlinear effects can easily be captured by time-domain models, but cannot be directly accomodated in a linear frequency-domain model. In QuLAF, the mooring system is represented by a linearized stiffness matrix for each wind speed, which is extracted from the SoA model and where hydrodynamic loads on the mooring lines are neglected. The dependency of the mooring matrix on wind speed is necessary because different mean wind speeds generally produce different mean thrust forces, which displace the floating wind turbine to different equilibrium states. The stiffness of the mooring system is different at each equilibrium position because of the nonlinear force-displacement behaviour of the catenary mooring lines.

For each wind speed, a first SoA simulation is needed with steady uniform wind and no waves, where only the tower fore-aft and platform surge, heave and pitch DoFs are enabled. After some time the floating wind turbine settles at its equilibrium position $(\xi_{eq,1}, \xi_{eq,3}, \xi_{eq,5})$, which is stored. These simulations should be just long enough so that the equilibrium state is reached (600 s in this case). Then, a new short SoA simulation with all DoFs disabled is run, where the platform initial position is the equilibrium with a small positive perturbation in surge, $(\xi_{eq,1} + \Delta\xi_1, \xi_{eq,3}, \xi_{eq,5})$. This simulation should be just long enough for the mooring lines to settle at rest (120 s in this case). The global mooring forces in surge and heave and the global mooring moment in pitch are stored, namely $(F_{moor,1}^{\xi1+}, F_{moor,3}^{\xi1+}, \tau_{moor,5}^{\xi1+})$. The process is repeated now with a negative




perturbation in surge $(\xi_{eq,1} - \Delta\xi_1, \xi_{eq,3}, \xi_{eq,5})$, giving $(F^{\xi1-}_{moor,1}, F^{\xi1-}_{moor,3}, \tau^{\xi1-}_{moor,5})$. All this information is enough to compute the first column of the mooring matrix $\mathbf{C}_{moor}$ for the wind speed $W$. Perturbations in heave $\Delta\xi_3$ and pitch $\Delta\xi_5$ provide the necessary information to compute the rest of the columns, and therefore the full matrix:

$$\mathbf{C}_{moor} = - \begin{bmatrix} \dfrac{F^{\xi1+}_{moor,1} - F^{\xi1-}_{moor,1}}{2\Delta\xi_1} & \dfrac{F^{\xi3+}_{moor,1} - F^{\xi3-}_{moor,1}}{2\Delta\xi_3} & \dfrac{F^{\xi5+}_{moor,1} - F^{\xi5-}_{moor,1}}{2\Delta\xi_5} & 0 \\ \dfrac{F^{\xi1+}_{moor,3} - F^{\xi1-}_{moor,3}}{2\Delta\xi_1} & \dfrac{F^{\xi3+}_{moor,3} - F^{\xi3-}_{moor,3}}{2\Delta\xi_3} & \dfrac{F^{\xi5+}_{moor,3} - F^{\xi5-}_{moor,3}}{2\Delta\xi_5} & 0 \\ \dfrac{\tau^{\xi1+}_{moor,5} - \tau^{\xi1-}_{moor,5}}{2\Delta\xi_1} & \dfrac{\tau^{\xi3+}_{moor,5} - \tau^{\xi3-}_{moor,5}}{2\Delta\xi_3} & \dfrac{\tau^{\xi5+}_{moor,5} - \tau^{\xi5-}_{moor,5}}{2\Delta\xi_5} & 0 \\ 0 & 0 & 0 & 0 \end{bmatrix} \qquad (36)$$

The first element of the mooring matrix $C_{moor,11}$ is shown as a function of wind speed in Fig. 6. It is observed that the stiffness in surge reaches its maximum around rated wind speed ($11.4$ m/s), where the thrust is also maximum and the floating wind turbine is the furthest from its equilibrium position.

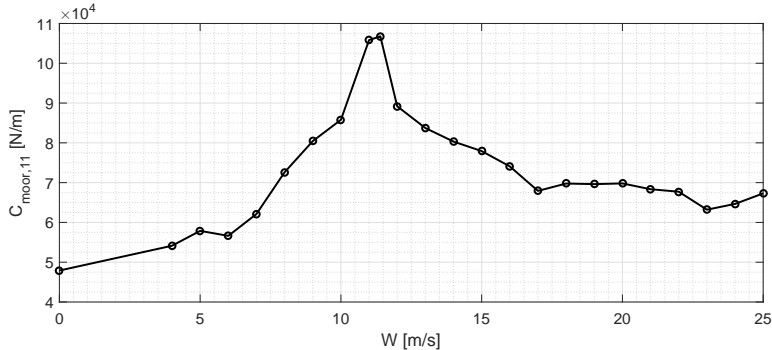

**Figure 6.** Surge mooring stiffness as a function of wind speed.

In the method applied here, the linearization of the mooring system has been done with the state-of-the-art model. However, in a real design study where the mooring characteristics change, the above procedure can be made significantly faster by direct
static analysis of the nonlinear mooring reactions around the floater equilibrium positions.

### 5.9   Estimation of extreme responses: a spectral approach

Classical Monte-Carlo analysis of response to stochastic loads entails running a simulation, extracting the peaks from the response time series, sorting them in ascending order and assigning an exceedance probability to each peak based on their position in the sorted list. Several simulations of the same environmental conditions with different random seeds provide a
cloud of points in the exceedance probability plot, from which an extreme response can be estimated. In this section, the linear nature of the simplified model will be further exploited to obtain an estimation of the extreme responses to wave loads by solely using the wave spectrum and the system transfer function, thus eliminating the need of a response time series and the



bias introduced by a particular random seed. An extension of the method to wind and wave forcing is further presented and discussed.

In a Gaussian, narrow-banded process, the peaks follow a Rayleigh distribution. In stochastic sea states, the linear free-surface elevation $\eta(t)$ is a Gaussian random variable $R_\eta$ with zero mean, thus the wave crests follow a Rayleigh distribution

(Longuet-Higgins, 1956)

$$P(R_\eta > \eta) = \exp\left\{-\frac{1}{2}\left(\frac{\eta}{\sigma_\eta}\right)^2\right\}, \tag{37}$$

where the variance of $\eta(t)$ is $\sigma_\eta^2$, which can be obtained from the integral of the wave spectrum,

$$\sigma_\eta^2 = \int_0^\infty S_\eta(\omega)d\omega. \tag{38}$$

If we consider only linear wave forcing, for the linear system in Eq. (3) the response is also Gaussian. If the response is

also narrow-banded, its exceedance probability can be found via the standard deviation of the response, which in turn can be obtained by integration of the response spectrum. From Eq. (3) we have

$$\hat{\boldsymbol{\xi}}(\omega) = \mathbf{H}(\omega)\hat{\mathbf{X}}(\omega)\hat{\eta}(\omega) \quad \implies \quad \hat{\boldsymbol{\xi}}_{glob}(\omega) = \mathbf{T}_{glob}\mathbf{H}(\omega)\hat{\mathbf{X}}(\omega)\hat{\eta}(\omega) \equiv \mathbf{TF}_{\eta\to\xi}(\omega)\hat{\eta}(\omega), \tag{39}$$

where $\mathbf{TF}_{\eta\to\xi}(\omega)$ is a direct transfer function from surface elevation to global response. The global response spectra $\mathbf{S}_{\xi,glob}(\omega)$ is related to the wave spectrum $S_\eta(\omega)$ in a similar way (Naess and Moan, 2013),

$$\mathbf{S}_{\xi,glob}(\omega) = \mathbf{TF}_{\eta\to\xi}(\omega)S_\eta(\omega)\mathbf{TF}_{\eta\to\xi}^{*T}(\omega). \tag{40}$$

Here $\mathbf{D}^{*T}$ indicates the transpose and complex conjugate of $\mathbf{D}$. By virtue of Eq. (37), the exceedance probability of e.g. the surge response $\xi_1$ is known from the the variance of the surge response $\sigma_{\xi,1}^2$, which is given by

$$\sigma_{\xi,1}^2 = \int_0^\infty S_{\xi,glob,11}(\omega)d\omega. \tag{41}$$

For nacelle acceleration, we can write the response as a function of the global nacelle displacement $\xi_{glob,4}$, therefore

$$\hat{\ddot{\xi}}_{glob,4}(\omega) = -\omega^2\hat{\xi}_{glob,4}(\omega) \quad \implies \quad \sigma_{\xi,4}^2 = \int_0^\infty \omega^4 S_{\xi,glob,44}(\omega)d\omega. \tag{42}$$



The turbulent part of the wind speed can also be considered a Gaussian random variable (Longuet-Higgins, 1956). On the other hand, aerodynamic loads are not a linear function of wind speed. Therefore the response to wind loads cannot be assumed to be Gaussian, and the approach shown above is not valid. However, the method above can be applied to cases with wind and wave forcing, bearing in mind that the necessary assumptions are not fulfilled, and therefore the results may not be accurate. If

wind and wave forcing are considered, Eq. (3) can be written as

$$\hat{\boldsymbol{\xi}}(\omega) = \mathbf{H}(\omega)\hat{\mathbf{F}}(\omega) \quad \implies \quad \hat{\boldsymbol{\xi}}_{glob}(\omega) = \mathbf{T}_{glob}\mathbf{H}(\omega)\hat{\mathbf{F}}(\omega) \equiv \mathbf{TF}_{F\to\xi}(\omega)\hat{\mathbf{F}}(\omega), \tag{43}$$

where $\mathbf{TF}_{F\to\xi}(\omega)$ is a direct transfer function from load to global response. The global response spectra $\mathbf{S}_{\xi,glob}(\omega)$ is now given by (Naess and Moan, 2013)

$$\mathbf{S}_{\xi,glob}(\omega) = \mathbf{TF}_{F\to\xi}(\omega)\mathbf{S}_F(\omega)\mathbf{TF}_{F\to\xi}^{*T}(\omega). \tag{44}$$

Here $\mathbf{S}_F(\omega)$ is the spectra of the total loads (hydrodynamic and aerodynamic),

$$\mathbf{S}_F(\omega) = \frac{1}{2d\omega}\hat{\mathbf{F}}(\omega)\hat{\mathbf{F}}^{*T}(\omega). \tag{45}$$

This method provides the exceedance probability of the dynamic part of the response, therefore the static part should be added *after* applying Eq. (37). Exceedance probability results from this method are compared in the next section to the traditional way of peak extraction from response time series.

**6    Validation of the QuLAF model**

We now compare and discuss the QuLAF and FAST responses to the same environmental conditions (see Tab. 4) representative of the Gulf of Maine (Krieger et al., 2015). The cases considered include irregular waves with and without turbulent wind. In all cases, the total simulated time was 5400 s in both models. The first 1800 s were neglected to discard transient effects in the time-domain model. The surface elevation of irregular sea states was computed in FAST from a Pierson-Moskowitz

spectrum, and the turbulent wind fields in TurbSim from an IEC-Kaimal spectrum. Since the turbulent wind fields used in the SoA simulations are the same employed for the precomputation of aerodynamic loads, and the free-surface elevation signal in the cascaded model is also taken from the FAST simulation, a deterministic comparison of time series is possible for all cases. In the plots shown in this section, the left-hand side shows a portion of the time series of wind speed at hub height, free-surface elevation, platform surge, heave and pitch, and nacelle acceleration; and the right-hand side shows PSD of the same signals.

The PSD signals were smoothened with a moving-average filter of 20 points to ease the spectral comparison between models. The blue vertical lines in the PSD plots indicate the position of the system natural frequencies predicted by the simplified model (see Tab. 3). In addition, exceedance probability plots of the responses with both models are shown, based on peaks



extracted from time series. The peaks were sorted and assigned an exceedance probability based on their position in the sorted list. The exceedance probability of the extracted peaks is compared to the one estimated with the method described in section 5.9 — these curves are labeled as "Rayleigh".

## 6.1 System identification

The system natural frequencies were calculated in QuLAF by solving the eigenvalue problem in Eq. (20). In FAST, decay simulations were carried out with all DoFs active, where an initial displacement was introduced in each relevant DoF and the system was left to decay. A PSD of the relevant response revealed the natural frequency of each DoF. A comparison of natural frequencies and periods found with the two models is given in Tab. 3, where it is shown that all platform natural frequencies in the model are within 1.3% error compared to the SoA model. On the other hand, the tower frequency is 8.6 % below the one
estimated in FAST. This difference is due to the absence of flexible blades in the simplified model, which are known to affect the coupled tower natural frequency. With rigid blades, the SoA model predicts a coupled tower natural frequency of 0.684 Hz, only 0.3 % above the tower frequency in QuLAF.

**Table 3.** Natural frequencies and periods obtained in FAST and QuLAF.

|  | Surge | Heave | Pitch | Tower |
|---|---|---|---|---|
| Natural frequency FAST [Hz] | 0.0054 | 0.0478 | 0.0316 | 0.746 |
| Natural frequency QuLAF [Hz] | 0.0054 | 0.0480 | 0.0320 | 0.682 |
| Error [%] | 0.00 | +0.42 | +1.27 | -8.58 |
| Natural period FAST [s] | 185.19 | 20.92 | 31.65 | 1.34 |
| Natural period QuLAF [s] | 185.19 | 20.83 | 31.25 | 1.47 |
| Error [%] | 0.00 | -0.42 | -1.25 | +9.38 |

The model presented here can be calibrated against other numerical or physical models if needed, by introducing user-defined additional restoring and damping matrices. For the present study, however, no calibration against the state-of-the-art
model was applied, in order to keep the model calibration-free and assess its suitability for optimization loops.

## 6.2 Response to irregular waves

The response to irregular waves with $H_s = 6.14$ m and $T_p = 12.5$ s (case "Waves 5" in Tab. 4) is shown in Fig. 7. On the PSD side, all motions show response mainly at the wave frequency range, and there is a very good agreement between both models for surge and heave. In pitch — and consequently in nacelle acceleration — the QuLAF model shows a lower level of
excitation at the wave frequency range when compared to FAST. This deviation was traced to the absence of viscous forcing in the simplified model, since the two pitch responses are almost identical if viscous effects are disabled in both models. As expected, the agreement is better for milder sea states, where viscous forcing is less important. In surge and pitch, some energy



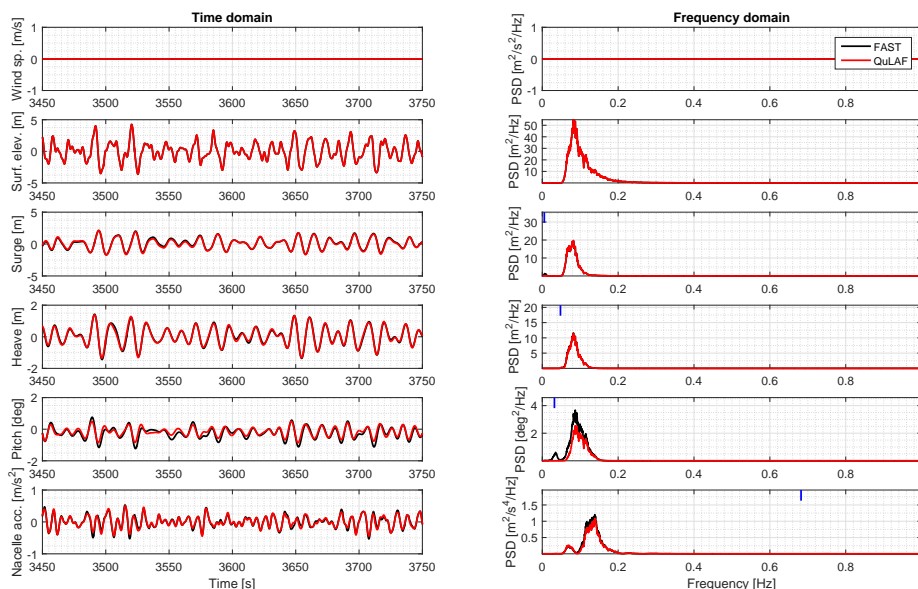

**Figure 7.** Response to irregular waves in time and frequency domain.

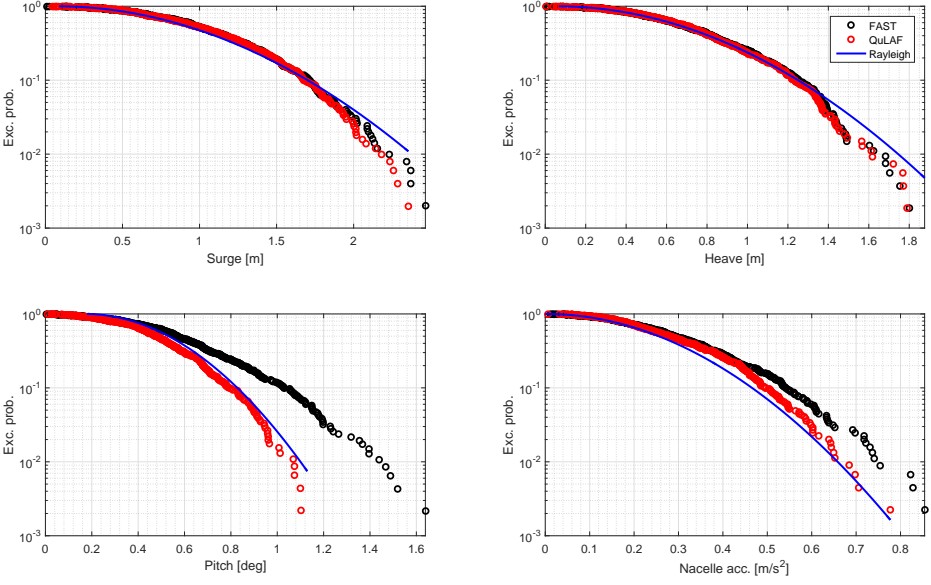

**Figure 8.** Exceedance probability of the response to irregular waves.





is visible at the natural frequencies, only in the FAST model. Since the peaks lie out of the wave spectrum and are not captured by QuLAF, they could originate from nonlinear mooring effects or from the drag loads, which are also nonlinear.

Figure 8 shows exceedance probability plots of the response to irregular waves. The Rayleigh curves fit well the responses given by the simplified model, which is expected, given that the free-surface elevation and the hydrodynamic forcing are

linear in the model, and the response can be considered narrow-banded. In the comparison between the two models, the surge and heave peaks are very well estimated by QuLAF. In nacelle acceleration and especially in pitch, however, the model underpredicts the response, with a difference of about 30% in pitch and about 8% in nacelle acceleration for the largest peak when compared to FAST. These observations in extreme response are consistent with the spectral results of Fig. 7 discussed above.

**6.3  Response to irregular waves and turbulent wind**

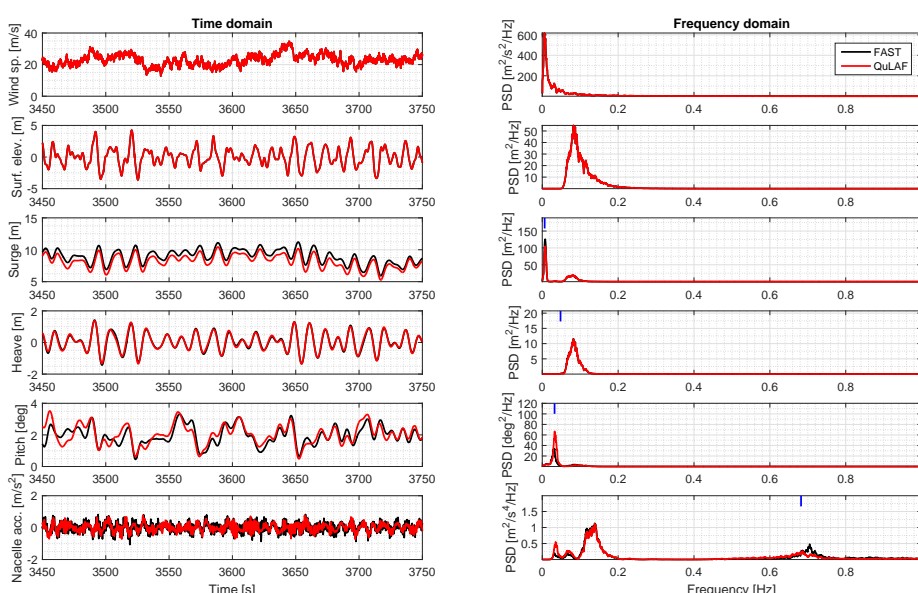

**Figure 9.** Response to irregular waves and turbulent wind in time and frequency domain.

The response to irregular waves with $H_s = 6.14$ m and $T_p = 12.5$ s (case "Waves + wind 5" in Tab. 4) and turbulent wind at $W = 22$ m/s is shown in Fig. 9. The surge motion is dominated by the surge natural frequency, which is clearly excited by the wind forcing. The linear model slightly underpredicts this resonance of the wind forcing with the surge natural frequency. Heave is dominated by the wave forcing, and the response of both models agree. In pitch, resonance with the natural frequency

also exists in both models, although QuLAF predicts more energy at that frequency than FAST. This overprediction also leaves



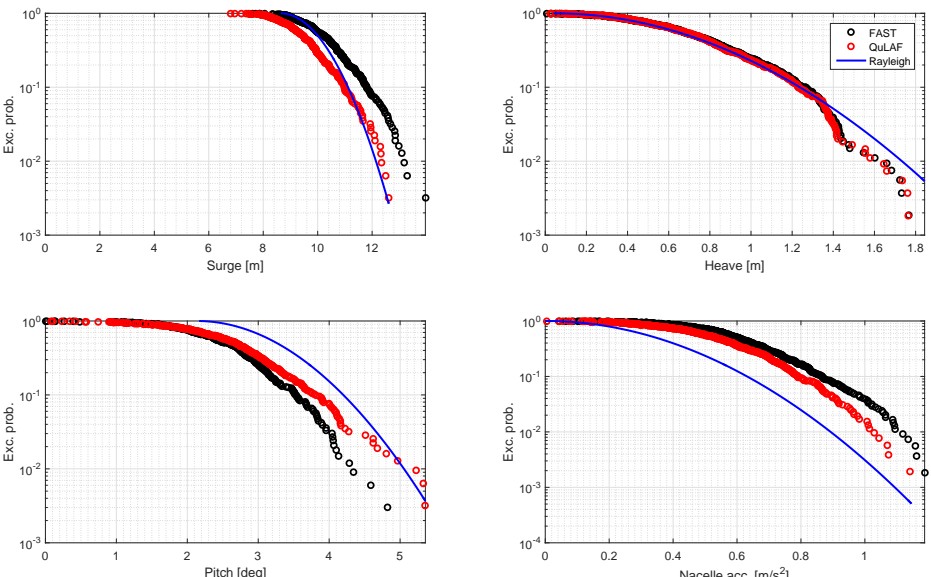

**Figure 10.** Exceedance probability of the response to irregular waves and turbulent wind.

a footprint on the PSD of nacelle acceleration, which shows energy at the pitch natural frequency, the wave frequency range and the tower natural frequency. The tower mode at 0.682 Hz, however, is slightly underpredicted by QuLAF.

The associated exceedance probability plots are shown in Fig. 10. In this case, the Rayleigh curves generally do not fit the responses predicted by the linear model, because the extreme peaks are no longer Rayleigh-distributed. This is because the
5 nonlinear nature of the wind loads makes the response non-Gaussian, and in some cases broad-banded with distinct frequency bands excited — i.e. the tower response cannot be considered narrow-banded here. The best fit is seen for heave, which is mainly excited by wave loads. When compared to FAST, however, QuLAF shows a good agreement with errors in the largest response peaks of approximately 8% in surge, 12% in pitch and 4% in nacelle acceleration.

### 6.4   Comparison of fatigue damage-equivalent loads

Table 4 below shows a summary of fatigue DELs for a wider range of environmental conditions. Each case is defined by the significant wave height $H_s$, the wave peak period $T_p$ and the mean wind speed $W$. The fatigue damage-equivalent bending moment at the tower bottom estimated with the two models is presented, as well as the error for the simplified model. Finally, the last column shows the ratio between the simulated time and the CPU time in QuLAF, $T_{rel}$. The cases labeled as "5" correspond to the results discussed in the previous section. The two DEL columns in Tab. 4 are also shown in Fig. 11 as a bar
plot.



**Table 4.** Summary of environmental conditions Krieger et al. (2015) and DEL results obtained in FAST and QuLAF.

| Case | $H_s$ [m] | $T_p$ [s] | $W$ [m/s] | $DEL_{FAST}$ [MNm] | $DEL_{QuLAF}$ [MNm] | $Error$ [%] | $T_{rel}$ [−] |
|---|---|---|---|---|---|---|---|
| Waves 1 | 1.51 | 7.65 | - | 75.69 | 76.44 | +1.00 | 2402 |
| Waves 2 | 1.97 | 8.00 | - | 98.44 | 98.62 | +0.19 | 2695 |
| Waves 3 | 2.43 | 8.29 | - | 120.74 | 119.95 | -0.65 | 2595 |
| Waves 4 | 3.97 | 9.85 | - | 179.45 | 170.55 | -4.96 | 2404 |
| Waves 5 (Figs. 7, 8) | 6.14 | 12.50 | - | 219.31 | 194.63 | -11.25 | 2595 |
| Waves + wind 1 | 1.51 | 7.65 | 6.0 | 167.13 | 158.74 | -5.02 | 1354 |
| Waves + wind 2 | 1.97 | 8.00 | 9.0 | 290.96 | 284.53 | -2.21 | 1409 |
| Waves + wind 3 | 2.43 | 8.29 | 11.4 | 375.12 | 349.37 | -6.87 | 1400 |
| Waves + wind 4 | 3.97 | 9.85 | 17.0 | 319.95 | 324.68 | +1.48 | 1365 |
| Waves + wind 5 (Figs. 9, 10) | 6.14 | 12.50 | 22.0 | 339.01 | 348.77 | +2.88 | 1408 |

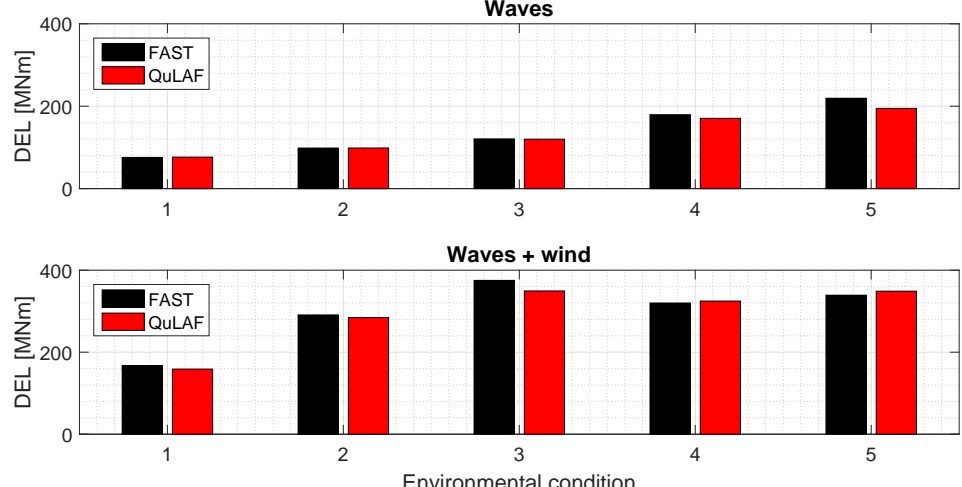

**Figure 11.** Damage-equivalent bending moment at the tower bottom for different environmental conditions.

For the cases with waves only, the model underpredicts the DEL at the tower bottom with errors from 0.2% to 11.3% that increase with the sea state, as observed in Fig. 11. The significant wave height also increases with the sea state, as do the associated nonlinear effects of position-dependent mooring stiffness and viscous hydrodynamic forcing, which are both included in FAST. As a linear model, QuLAF's accuracy is bounded to the assumptions of small displacements around the equilibrium point. Hence, it is expected that the linear model performs worse for the environmental conditions where nonlinear effects are not negligible. This observation is also consistent with the discussion around Fig. 7, which corresponds to the most severe sea state considered here.



For the cases with wind the errors range from 1.5% to 6.9%, but the trend is not as clear. The predictions seem to be worst for the environmental condition corresponding to rated wind speed. Around rated speed the wind turbine operation switches between the partial- and the full-load regions, which correspond to very distinct regimes of the generator torque and blade pitch controller. The complexity of the dynamics involved in this transition zone is not well captured by the simplified model. The

vibration of the tower is also more likely to be excited around rated wind speed, when the thrust is maximum. As the coupled tower natural frequency is different for the two models, this will also have an impact on the resulting DEL. On the other hand, the aerodynamic simplifications in the cascaded model seem to work best for wind speeds above rated. The last column of Tab. 4 shows that the ratio between simulated time and CPU time is between 1300 and 2700 in a standard laptop with an Intel Core i5-5300U processor at 2.30 GHz and 16 GB of RAM. In other words, all the simulations in Tab. 4 together, 1.5 h long each,

can be done in about half a minute.

## 7    Conclusions

A model for Quick Load Analysis of Floating wind turbines, QuLAF, has been presented and validated. The model is a linear, frequency-domain tool with four planar degrees of freedom: platform surge, heave, pitch and tower modal deflection. The model relies on higher-fidelity tools from which hydrodynamic, aerodynamic and mooring loads are extracted and cascaded.

Hydrodynamic and aerodynamic loads are precomputed in WAMIT and FAST respectively, while the mooring system is linearized around the equilibrium position for each wind speed using MoorDyn. A simplified approach for viscous hydrodynamic damping was implemented, and the decay-based extraction of aerodynamic damping of Schløer et al. (2018) was extended to multiples degrees of freedom. Without introducing any calibration, a case study with a semi-submersible 10MW configuration showed that the model is able to predict the motions of the system in stochastic wind and waves with acceptable accuracy.

The damage-equivalent bending moment at the tower bottom is estimated with errors between 0.2% and 11.3% for all the five load cases considered in this study, covering the operational wind speed range. The largest errors were observed for the most severe wave climates in wave-only conditions and for turbine operation around rated wind speed for combined wind and wave conditions. The computational speed in QuLAF is between 1300 and 2700 times faster than real-time. Although not done in this study, introducing viscous hydrodynamic forcing and calibration of the damping against the SoA model would likely result

in improved accuracy, but at the expense of lower CPU efficiency and less generality in the model formulation.

It has been shown that the model can be used as a tool to explore the design space in the preliminary design stages of a floating platform for offshore wind. The model can quickly give an estimate of the main natural frequencies, response and loads for a wide range of environmental conditions, which makes it useful for optimization loops. Although a better performance may be achieved through calibration, a calibration-free approach was used here to emulate the reality of an optimization loop within

the design process. In such process, once a potentially optimal design has been found, a full aero-hydro-servo-elastic model is still necessary to assess the performance in a wider range of environmental conditions, including nonlinearities, transient effects and real-time control. Since the model is directly extracted from such a state-of-the-art model, this step can readily





be taken. While the state-of-the-art model should thus still be used in the design verification, the present model provides an efficient and relatively accurate complementary tool for rational Engineering design of offshore wind turbine floaters.

*Competing interests.* The authors declare that they have no conflict of interest.

*Acknowledgements.* This work is part of the LIFES50+ project (www.lifes50plus.eu). The research leading to these results has received

5  funding from the European Union's Horizon 2020 research and innovation programme under grant agreement No. 640741.



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
