# Peer review of "An efficient frequency-domain model for quick load analysis of floating offshore wind turbines"

_Wind Energy Science, 2018_

## Referee Comment (RC1) · Anonymous Referee #1 · 4 May 2018

This paper presents a simplified modeling approach, called QuLAF, to calculate tower-base loads in a floating wind turbine. The approach is an interesting one and is well thought out and presented.

Some items that I think would make the paper better include a larger discussion on what makes this modeling approach unique from others that have done simplified modeling in the past. Other work is presented, but the differences are not well described.

A second point would be to better describe how the authors see this approach benefiting the design process for a floating wind turbine. There appears to be several steps in developing the simplified model which could make it time consuming, such as the

extraction of damping coefficients. How much of this work can be automated, versus how much needs to be done manually? What would the total time to develop this approximated model from the original? With super computers now, 50,000 simulation could be run in a couple of days. In addition, the authors are still using WAMIT in the pre-computation stage, which will be time consuming. The time savings seems to come from being able to do multiple simulations for the same design. However, it does not seem like this approach would allow designers to quickly examine different design approaches due to the time components for creating the model.

Why not consider using a Morison model for the hydrodynamic loading? While it may not be completely accurate for larger structures, it seems the represent the system fairly well, especially considering the level of accuracy in this simplified approach. Was a comparison to this approach done?

While I can see such a model could predict steady-state loading, and thus be able to estimate the fatigue loading of the system, it would not capture the discrete events that tend to cause extreme loading in the system, which can be a design driver. I therefore think a more thorough discussion of where this tool fits within the design process would be beneficial.

---

## Referee Comment (RC2) · T. A. Nygaard (Referee) · 23 Jun 2018

The main review comments are in: comments2_nygaard_wes2018_25.pdf

Minor comments, questions and edits for consideration are stated directly in the article with highlights and sticky notes in: wes-2018-25_comments_3.pdf

Please also note the supplement to this comment:
https://www.wind-energ-sci-discuss.net/wes-2018-25/wes-2018-25-RC2-supplement.zip

---

## Author Response (AR1)

**Referee 1 - Anonymous**

The authors thank the referee for the feedback provided. Please find below the referee's comments (RC), the corresponding author's comments (AC) and the changes in the manuscript. PXLY refers to page X and line Y in the *revised* manuscript.

RC: This paper presents a simplified modeling approach, called QuLAF, to calculate towerbase loads in a floating wind turbine. The approach is an interesting one and is well thought out and presented.

RC: Some items that I think would make the paper better include a larger discussion on what makes this modeling approach unique from others that have done simplified modeling in the past. Other work is presented, but the differences are not well described.

AC: To the authors' knowledge, this work is the first simplified tool for floating wind turbines to include both stochastic wind and waves (see P2L24), and to compare not only motion PSDs but also extreme values and fatigue loads. This has been made clearer in the end of Section 1, see for example P3L17.

RC: A second point would be to better describe how the authors see this approach benefiting the design process for a floating wind turbine.

AC: The model is meant to complement existing state-of-the-art tools, giving a preliminary quick overview of the response and loads for a wide range of environmental conditions. After this preliminary screening, the time-domain model should be used to analyze in more detail specific load cases - e.g. cases with extreme loads or transient events (see P3L22-24 and P32L16-22). See also new Section 5.10.

RC: There appears to be several steps in developing the simplified model which could make it time consuming, such as the extraction of damping coefficients. How much of this work can be automated, versus how much needs to be done manually? What would the total time to develop this approximated model from the original? With super computers now, 50,000 simulation could be run in a couple of days.

AC: For this study, the focus has been on assessing the simplified approach and identifying potential improvements, therefore many things have been done manually (e.g. linearization of mooring system and extraction of aerodynamic damping). However, the authors believe that most of this work can be automated if needed. In addition, aerodynamic loads and damping coefficients have to be extracted only once for a given wind turbine. We cannot give an exact figure on the time spent developing the model because it has been an incremental process, through which we have tried many ideas that finally were not included in the final version. It is true that supercomputers make state-of-the-art models more attractive, but not all concept developers have access to such resources and the simplified model will always run a few order of magnitude faster (e.g. the 50,000 simulations with QuLAF would take a couple of minutes in a supercomputer). This discussion has been added to the manuscript, see new Section 5.10.

RC: In addition, the authors are still using WAMIT in the pre-computation stage, which will be time consuming. The time savings seems to come from being able to do multiple simulations for the same design. However, it does not seem like this approach would allow designers to quickly examine different design approaches due to the time components for creating the model. Why not consider using a Morison model for the hydrodynamic loading? While it may not be completely accurate for larger structures, it seems the represent the system fairly well, especially considering the level of accuracy in this simplified approach. Was a comparison to this approach done?

AC: The choice of a radiation-diffraction solver for the hydrodynamic modelling was motivated by the study case, given the shape and size of the chosen floating substructure. In an optimization process where many design variations are to be evaluated, the WAMIT panel geometry can still be parameterized and the WAMIT analysis can be done automatically. On the other hand, for slender simpler geometries (such as spars) it would be natural to employ a Morison approach, thus simplifying the whole process. No comparison to the Morison approach has been done in this study. This discussion has been added to the manuscript, see new Section 5.10.

RC: While I can see such a model could predict steady-state loading, and thus be able to estimate the fatigue loading of the system, it would not capture the discrete events that tend to cause extreme loading in the system, which can be a design driver. I therefore think a more thorough discussion of where this tool fits within the design process would be beneficial.

AC: As stated in the paper, the model presented here is not meant to replace state-of-the-art tools, but rather to complement them by allowing a faster exploration of the design space. In addition, the QuLAF and FAST models presented in this study have been recently used in the LIFES50+ project for a broader analysis of different design-driving load cases, including normal operation, extreme and transient events (report available at http://lifes50plus.eu/wp-content/uploads/2018/07/D78-GA_640741.pdf). Generally, the results were quite satisfactory and the main findings and model limitations are in line with the ones discussed in the paper. In the extended study the effect of aerodynamic damping on tower vibrations was found to also play an important role in the DEL prediction. The extended report is now mentioned in the paper and the discussion has been extended in Section 7, including the effect of aerodynamic damping on tower vibrations.
* * *
Please note that other minor changes have been introduced in the text to improve readability and fix a few typos. Figure 3 has also been improved.

**Referee 2 – Tor A. Nygaard**

The authors thank the referee for the feedback provided. Please find below the referee's comments (RC), the corresponding author's comments (AC) and the changes in the manuscript. PXLY refers to page X and line Y in the *revised* manuscript.

RC: I enjoyed reading this article. It is easy to read, has a complete set of equations, and explains the results very well.

RC: This work is relevant. Floating wind turbine evaluations with State-of-the-Art (SoA) time-domain integrated models require significant resources in terms of computations and post-processing of the results. The load case matrix is large, and usually each load case is computed with several realizations of irregular waves and turbulent wind. In floating wind turbine research, the focus has mostly been on time-domain models, due to concern about large motions, nonlinearities and coupling. As these models mature, and experience is gained with different floating platforms, it seems like many cases can be properly linearized and solved in the frequency domain. The impact of this work could be extension of time-domain integrated models to allow efficient computations of some of the load cases in the frequency domain. I think the key to application of methods like the one presented in this article (QuLAF) is automation of the input. If a SoA model is set up for input preparation to QuLAF, the choice is then to just run the SoA model for all the load cases by cloud computing, or invest in some additional work setting up QuLAF, which hopefully then will be recovered by the very fast execution of QuLAF.

AC: Agree. Although the focus of this study has been on assessing the simplified approach and identifying potential improvements, and therefore many things have been done manually (e.g. linearization of mooring system and extraction of aerodynamic damping), the authors believe that most of this work can be automatized if needed. This discussion has been added to the manuscript, see new Section 5.10.

RC: The quality of the article is very good. In my opinion, it lacks only a few clarifications to be ready for publication.

RC: The description of QuLAF, section 5 is quite complete, but I think it would benefit from a few statements right away, on the forcing term on the right-hand-side (RHS) of eq. 5. This information is given later in the paper, but it would be easier to understand the mass matrix, eq. 4, with this information upfront.

AC: As suggested, Sections 5.1 Dynamic response vector and 5.2 Dynamic load vector have been moved to the beginning of Section 5, right after the equation of motion (eq.3) and before the matrices are introduced.

RC: From eq. 15, we can see that the external forces are transformed to forces and moments at the water line, component 1 -3 in the RHS F. The physical interpretation of component 4 is not mentioned in the article; to me it looks like it represents the part of the external force/moment (component 1 -3) performing work on tower deflection.

AC: That is correct, the last component of Faero represents the effect of aerodynamic loads on the tower modal deflection, and includes aerodynamic thrust force and tilt torque at the shaft. This information has now been added to the text at the end of Section 5.2.

RC: Instead of just defining the mass matrix, I suggest a few sentences on how it is derived (energy method?). All components of the mass matrix except (4,4) can be understood directly by looking at which

forces are required to produce unit accelerations along DOF 1 -3. For example, column 1 (and row 1) is the forcing required to produce a unit horizontal acceleration, with no tower bending. Column 4 represents the external (component 1 -3, already known from symmetry) and internal (component 4) forcing required to obtain a tower top acceleration of phi_hub.

AC: The matrices were derived from a free body diagram where all the forces were included using D'Alembert principle. More precisely, the mass matrix it was derived by looking at the forces needed to produce unit accelerations in each DoF. A note on this was added to the text in the beginning of Section 5.3.

RC: Consider moving the sections 5.4, dynamic response vector and 5.5, dynamic load vector to the beginning of section 5; this would probably solve the issues mentioned above.

AC: As suggested, Sections 5.1 Dynamic response vector and 5.2 Dynamic load vector have been moved to the beginning of Section 5, right after the equation of motion (eq.3) and before the matrices are introduced.

RC: For a floating wind turbine with a catenary mooring system, mean drift and current can be important for the mooring line characteristics at the mean platform position. The way I understand the model, this can be taken into account when evaluating mooring line and other mean position characteristics with the SoA model. If this is the case, I suggest mentioning explicitly that mean drift (along the wave direction) and current from any direction can be taken into account in QuLAF.

AC: True. Although only wind has been considered in this paper, the position-dependent mooring stiffness matrix can include effects from other mean forces such as mean drift and current. This has now been stated in the text at the end of Section 5.5.

RC: Misalignment of wind, waves and current can be important for fatigue calculations. I think the article would benefit from a few comments on extension of QuLAF to include sway and roll. Do the authors think this would be straightforward, or are there issues with coupling terms etc.?

AC: The extension of QuLAF to out-of-plane degrees of freedom is on the list of possible improvements. The authors do not foresee major issues in doing so, and perhaps the aerodynamic loads and damping is where one should be more careful. A paragraph on future improvements has been added to the end of Section 7, including this and other possible improvements.

RC: A separate file contains the article, with highlights in yellow and sticky notes with minor questions/comments and edits for consideration.

AC: The suggested text edits have been implemented, and the questions/comments are addressed below.

P5L17: Although this section mainly serves as motivation, I think the mathematically straightforward switch from time to frequency domain deserves a more precise comparison than 'practically identical'. After the initial transient, the differences, if any, should be due to the finite time step in the time stepping scheme, and the corresponding finite number of frequencies in the  FFT/iFFT, right?  I assume you selected the time step based on a sensitivity study. For example, how much do the maximum deflections computed with the time domain approach change when doubling the time step ? How much do the corresponding solutions in the frequency domain change?. What is the maximum difference between deflections computed with time and frequency domain models differ (making sure the comparison is done after the initial transient is gone) . The time step of 0.01s is typical for time domain-simulations of full-scale floating

wind turbines. Does the selected values for mass, damping and stiffness in this example reflect one of the DOFs for the platform in this paper?

AC: Yes, the error between time- and frequency-domain solutions is mainly due to discretization. In the draft version, the example presented in Section 3 was obtained with a 1-DoF model of a lab-scale spar subjected to linear hydrodynamic forcing, used for teaching. Since the only purpose of this section is to illustrate the two methods to solve the equation of motion and to compare the execution time for the same time step, there is no relation between the properties of that spar and the semisub used in the paper. The time step in QuLAF is the same as in the SoA model, which was chosen based on a sensitivity study.

In the revised version, the example in Section 3 has been replaced by a 1-DoF model of the OC3-Hywind demo (full scale), and a second plot has been added to show how the error between time and frequency domain solutions behaves with time. The error has also been quantified and is mentioned in the text.

P10L16: is the nacelle total velocity (caused by platform surge, pitch and tower deflection) taken into account when computing the aerodynamic damping?

AC: The aerodynamic damping for each DoF (surge, pitch, tower) is extracted from a separate simulation where only the relevant DoF is active and all the other DoFs are restrained (see P18L28, P19L6 and the caption of Figure 4).

P23: This is small in a printed version

AC: All results plots have been trimmed and enlarged to improve readability.

P25L4: Gumbel distributions are often used in extreme value statistics. Would that be relevant here?

AC: We believe that in this case the Rayleigh distribution is more adequate to predict the distribution of peaks in one realization. Gumbel distributions, on the other hand, are useful to predict extreme values of many realizations (e.g. estimation of 50-year wave height from annual maxima).

P27L6: Did you try running FAST with rigid blades here? That would be an interesting comparison

AC: Yes, and for the case "Waves + wind 3" (the case with "worst" results in wind and waves) the DEL error changes from -6.87% to -5.56% when the blades are rigid in FAST, hence the blade flexibility plays a role here, but it is not the only cause. This discussion has been added to Section 6.4.

P27L7: why? is it more linear in this region than below rated ?

AC: We believe it has to do with the thrust curve being more "flat" above rated than below. This comment has been added to the end of Section 6.4.

P29: ok with web address as reference?

AC: We prefer to reference the software's website rather than a specific version of the manual. The same is done with FAST, MoorDyn, etc.
* * *
Please note that other minor changes have been introduced in the text to improve readability and fix a few typos. Figure 3 has also been improved.

[revised manuscript text omitted]